# "With group antenatal care, pregnant women know they are not alone": The process evaluation of a group antenatal care intervention in Ghana

Ruth Zielinski[1]*, Vida Kukula[2], Veronica Apetorgbor[2], Elizabeth Awini[2], Cheryl Moyer[3], Georgina Badu-Gyan[2], John Williams[2], Nancy Lockhart[1], Jody Lori[1]

1 Department of Health Behavior and Biological Sciences, School of Nursing, University of Michigan, Ann Arbor, Michigan, United States of America, 2 Ghana Health Service, Dodowa Health Research Centre, Dodowa, Ghana, 3 Department of Learning Health Sciences, University of Michigan Medical School, University of Michigan, Ann Arbor, Michigan, United States of America

* ruthcnm@umich.edu

## Abstract

### Background

An essential component to improving maternal and newborn outcomes is antenatal care. A trial of group antenatal care was implemented in Ghana where 7 health care facilities were randomized to be intervention sites and 7 control sites continued traditional antenatal care. Group antenatal care, where 10–14 women with similar due dates meet together for visits, includes traditional components such as risk assessment with increased opportunity for education and peer support. The study aim was to assess and report the process evaluation of the implementation of a group antenatal care intervention.

### Methods

Process evaluation data were collected alongside intervention data and included both quantitative and qualitative data sources. Midwives at the health facilities which were randomized as intervention sites completed tracking logs to measure feasibility of the intervention. Research team members traveled to intervention sites where they conducted structured observations and completed fidelity and learning methods checklists to determine adherence to the model of group antenatal care delivery. In addition, midwives facilitating group antenatal care meetings were interviewed and focus groups were conducted with women participating in group antenatal care.

### Results

In the majority of cases, midwives facilitating group antenatal care completed all components of the meetings with fidelity, following best practices such as sitting with the group rather than standing. Across 7 intervention sites, 7 groups (622 pregnant women) were documented in the tracking logs and of these participants, the majority (74%) attended more

**Data Availability Statement:** The data is available on Deep Blue: https://deepblue.lib.umich.edu/data.

**Funding:** Author JL was Principal Investigator on the National Institutes of Health; Eunice Kennedy Shriver National Institute of Child Health & Human Development. Grant#: RO1HD096277 Authors did not receive a direct salary from the funders. This funding did provide a percentage of effort for University of Michigan faculty and staff/authors as well as a percentage of effort for Dodawa Health Research Centre staff/authors. The funders had no role in study design, data collection and analysis, decision to publish, or preparation of the manuscript.

**Competing interests:** The authors have declared that no competing interests exist.

than half of the meetings, with 32% attending all 8 group meetings. Three themes were identified in both the midwife interviews and focus groups with pregnant women: 1) information sharing, 2) sense of community, and 3) time management challenges. An additional theme emerged from the focus groups with the women: women who had already given birth noticed a disconnect between what they learned and treatment received during labor and birth.

## Conclusion

This process evaluation determined that group antenatal care can be implemented with fidelity in low and middle-income countries. Time management was the biggest challenge, however both midwives and pregnant women found the model of care not only acceptable, but preferable to traditional care. The knowledge shared and sense of community built during the meetings was a valuable addition to the individual model of antenatal care.

## Introduction

Maternal and infant health have long been at the forefront of global health initiatives. Through concerted efforts to improve maternal and newborn health, some progress has been made. In 1990, newborn deaths worldwide were 5 million and by 2019 that had decreased to 2.4 million [1]. During the same time period the maternal mortality ratio (MMR—number of maternal deaths during a given time period per 100,000 live births) reduced by 38% to 211 per 100,000 live births [2]. However, the MMR is still three times higher than the goal set by the United Nation's Sustainable Development Goal (SDG) of less than 70 maternal deaths per 100,000 live births by the year 2030 [3]. The newborn death rate NMR of 27/100 is more than twice as high as the goal set by the SDGs of less than 12 newborn deaths per 1000 live births by the year 2030 [3]. Moreover, there has not been consistent improvement across countries with 99% of maternal deaths occurring in low- and middle-income countries (LMICs) [4]. Sub-Saharan African countries alone account for 86% of maternal deaths with an MMR of 415/100,000 and NMR of 27/1000; bearing the brunt of the loss of lives among newborns and mothers [4]. While death rates for both mothers and newborns consistently trended downward between 2000 and 2019, there is preliminary evidence that progress has stalled or in some cases been negated by the global COVID-19 pandemic [5]. This underscores the importance of implementing measures to improve maternal and newborn health that can be sustainable through events such as pandemics.

Antenatal care (ANC) defined by the World Health Organization as "care provided by skilled health-care professionals to pregnant women and adolescent girls to ensure the best health conditions for both mother and baby during pregnancy" [6] is one approach to reducing negative outcomes for mothers and infants. Providing comprehensive ANC is challenging especially in resources constrained countries. Long waiting times, lack of continuity of care by caregivers, and hurried consultations due to heavy workload on health providers results in dissatisfaction and missed visits [7]. Traditionally, ANC involves individual, brief, regular visits with a maternity care provider that focuses on risk reduction of complications such as pre-eclampsia [8]. An innovative alternative to the traditional model of ANC is group care which places clinical assessment, education and social support into a group setting. These group meetings retain the risk assessment components of ANC care such as monitoring blood pressure while providing an opportunity for education, empowerment and support. In the group ANC model, women with similar estimated due dates are organized into groups that meet at

regular intervals during pregnancy. Meetings typically last 1–2 hours which allows time for more in-depth discussions and participation from group members. The groups remain stable throughout which offers the additional benefit of social support from the group.

A National Institute of Health funded study (Clinical Trial # NCT04033003) entitled "Group antenatal care to promote a healthy pregnancy and optimize maternal and newborn outcomes: A cluster randomized controlled trial" (GRAND) has been implemented in Ghana to determine the effectiveness of the group ANC model of care. This report focuses on the process evaluation of the implementation of the intervention (group ANC). The overarching goal of the GRAND study is to determine whether group ANC facilitated by trained midwives, utilizing a fixed group model of 10–14 pregnant women by gestational age for the duration of the pregnancy, improves outcomes when compared with participants in routine, individual antenatal care. The randomization occurred at the facility level whereby 7 facilities were randomized to routine ANC and 7 were randomized to offer group ANC to pregnant women seeking ANC at the facility. Primary outcomes include birth preparedness, complication readiness, increased care-seeking behavior, and improved maternal and newborn outcomes. Enrollment in the study and participating in group ANC was voluntary. The project used a Health Literacy Skills Framework and details of the framework and project protocol have been described elsewhere [9].

While program outcomes measure the success of an intervention, a process evaluation is important to understand why an intervention is, or is not, successful and to ensure that the outcomes seen are a result of the intervention, particularly when implementation occurs in different cultures and contexts [10]. Preliminary data indicates that group ANC is feasible and may result in better outcomes for mothers and newborns in lower resourced countries [11], however larger studies of this method of ANC delivery are needed, particularly in sub-Saharan Africa given the unacceptably high rates of maternal and neonatal mortality and morbidity [6]. To that end, we developed and implemented a process evaluation of the group ANC intervention at the onset of the GRAND study to monitor fidelity, or the extent to which the group ANC intervention was implemented as planned (Table 1) and identify issues and barriers in a timely manner [12].

The purpose of this process evaluation was to identify and document barriers and facilitators to implementation of the group ANC intervention. Using both quantitative and qualitative methods, we aimed to identify potential and actual influences on the quality and conduct of the intervention's operations, implementation, and antenatal care delivery. The data collected were used for formative and summative purposes as a continuous monitoring and problem-solving approach to oversee group ANC model fidelity.

## Methods

### Overview of the intervention

During the summer of 2019, research team members with expertise in group ANC from the United States met with research partners in Ghana to conduct a train-the-trainer course for a

**Table 1. Components of the GRAND—Group antenatal care model.**

• Facilitated group ANC meetings covering the WHO recommendations on antenatal care for a positive pregnancy experience [12]
• Groups of approximately 10–14 pregnant women of similar gestational age, conducted while sitting in a circle.
• Groups led by two midwives trained in group facilitation and participatory learning.
• Physical assessment and procedures conducted one-on-one with midwife before or after the group ANC meetings
• Self-assessment including blood pressure, weight and whether a woman has experienced any danger signs

selected group of midwives that would then serve as champions to train the midwives working at the intervention sites. The training was undertaken in the same manner in which the group ANC would be conducted; in small groups, in a circle, where discussion and participation were encouraged. The ten champion trainers then traveled to the districts where they were supported by members of the research team as they trained an additional 52 midwives as facilitators for group ANC. Each training session lasted 3 full days and included opportunities for each midwife to practice being the facilitator. Each participant was given a Facilitator Guide and a set of facilitation tools to use to prepare for and use during each group ANC meeting.

## Process evaluation design

Following the midwife facilitator training, the champion trainers as well as other members of the Ghana research team received additional training in the data collection methods for the process evaluation of group ANC including facilitation of focus groups among pregnant women participating in group ANC at the intervention sites and conducting interviews with the midwife facilitators of group ANC at those sites. The process evaluation included both qualitative and quantitative data sources (Table 2) Data collection began at the implementation of the group ANC meetings and continued for the next 15 months.

## Study setting

Fourteen health centers in eastern Ghana that provided antenatal care were included in the GRAND study. Only the 7 sites randomized for the group ANC intervention were included for collection of process evaluation data since the evaluation was of group ANC implementation. Research team members traveled to the health facility intervention sites to observe group ANC meetings (completing the Fidelity and Learning Methods Checklists), to conduct interviews with the midwife facilitators and facilitate focus groups with group ANC pregnant participants.

## Participants

**Midwife facilitators.** Midwife facilitators In order to assess fidelity to the model of group ANC and to determine if all content was covered, structured observations were conducted by research team members with midwife facilitators as participants. Inclusion criteria was; being a midwife who had completed training was facilitating group ANC meetings and desire to participate as participation was voluntary. During a group ANC meeting, the research team member observed using the Fidelity Checklist and Learning Methods Checklists described below. Following the group ANC meeting, the midwives participated in an interview with the research team member who used a semi-structured interview guide. Demographic data was not collected from participants and any identifiers in the transcripts were de-identified prior to analysis.

**Women participating in Group ANC.** For the outcome evaluation, data were collected from all women participating in group ANC with a corresponding number of participants at

**Table 2. Data sources for process evaluation.**

- Structured Observations
  - Fidelity Checklist
  - Learning Methods Checklist
- Interviews with Midwives
- Tracking Logs
- Focus Groups with Participants of Group ANC

the control groups sites. Outcome data collection is still underway. However, for the process evaluation, focus group data was collected from a subset (1–2 groups) of participants enrolled from each of the intervention arm sites of the GRAND study. Focus groups were not conducted with participants in routine ANC at the control group sites as the goal was to gather information regarding group ANC. While demographic and health data were collected for the outcome evaluation, participants in the focus groups were not identified as individual participants in the focus group transcripts. Any identifiers such as midwife or health facility names were removed from the transcripts prior to analysis.

## Data collection process

**Structured observations.**  A sample of group ANC meetings were observed for each midwife facilitator to monitor fidelity to the model. Beginning with the initial groups in fall 2019, and continuing at intervals throughout the study period, members of the Ghana research team who had also participated as champion trainers traveled to the intervention site health facilities to observe the facilitation of group ANC meetings. Observations occurred during one of the later (5, 6 or 7) group ANC meetings. Observations included whether content was delivered as intended, whether women were encouraged to actively participate in group discussions and activities, whether picture cards were used as written in the Facilitators Guide, and whether feedback was provided to participants during demonstrations. During these observations the research team members completed both the Fidelity Checklist Scale and the Learning Methods Checklist (S1 and S2 Checklists).

**The fidelity checklist.**  This scale consists of 7 items scored as "Always", "Sometimes" or "Never". The items identify whether or not the midwife facilitator is using the techniques taught to encourage group dynamics and discussion. For example, the midwife should sit in the group rather than stand when conducting the meeting. Additional space is available for comments at the bottom of the Fidelity Checklist (S1 Checklist).

**The learning methods checklist.**  This is a 16-item checklist that identifies whether or not the midwife follows the format spelled out in the group ANC Facilitators Guide. The format is intentionally prescriptive to ensure all topics are adequately covered and there is group discussion around the topics. For this checklist, observers were instructed to place a check mark if the item was observed. Additional space is available for comments (S2 Checklist).

**Interviews.**  A face-to-face semi-structured interview was completed with a midwife facilitator from each intervention site (N = 7) following the structured observation of the group ANC meeting The purpose was to explore the midwives' perceptions of which components of the intervention were successfully implemented, to identify barriers and facilitators to implementation and to give suggestions for improvement. No personal data such as demographic information was collected from the participants, all of whom were licensed midwives working in the healthcare sector in Ghana. Informed consent for the interviews was obtained from the midwife facilitators and participation was voluntary. The interviews were conducted in English by a member of the research team using the semi-structured interview guide (included in S2 File). The research team members doing the interviews were given training in qualitative data collection methods. The interviews were audio-recorded and transcribed verbatim by research assistants.

**Tracking logs.**  A brief form was completed by the midwife facilitator each time a group ANC meeting was held to track the date of the meeting and the number of participants from the group in attendance. This measured "dose", or amount of the intervention each participant received for analysis of outcomes as well as feasibility by determining if participants were able to attend all or most of the group ANC meetings (S2 Checklist).

**Focus groups.** For each intervention site at least one focus group discussion was held with pregnant women participating in group ANC. These focus groups, conducted by a member of the research team trained in qualitative data collection methods, were held at the end of the group ANC meeting and were voluntary; women were welcome to leave the group if they did not want to participate. Using an open-ended question guide (included in S2 File), the participants were encouraged to share their perception of group care and offer suggestions for improvement. The interviews were conducted in the local language, Twi, audio recorded and then translated and transcribed verbatim by research assistants who are fluent in both English and Twi. Key phrases that were difficult to translate were left intact, with the closest approximate meaning put in parentheses in the transcription.

## Data analysis

Quantitative data (Fidelity Checklist and Learning Methods Checklist) were collected by the research team members during the observation of the group ANC then entered into an Excel spreadsheet. Data were analyzed descriptively to assess whether certain items, midwives, or intervention sites were performing less well when compared to others. Tracking logs were completed by the midwife facilitators, collected by the research assistants and also entered into an Excel spreadsheet. Data were analyzed descriptively by how many group ANC meetings each participant attended.

To analyze the qualitative data from the focus groups and interviews, a qualitative comparison approach was used. Qualitative comparison is useful when more than one standpoint of a phenomenon is of interest [13]. In this case we were interested in comparing and contrasting the experience of group ANC for both the midwives and the pregnant participants. First, the research team immersed themselves in the data by reading and re-reading the transcripts. Midwive's responses in the interview transcripts were analyzed thematically for codes (recurring words or phrases) by one team member and the pregnant participants responses from the focus groups were analyzed in a similar manner by another team member. The codes were reviewed by the research team to developed themes separately for each of the data sources. Authors then met to compare and contrast the themes to identify similarities and differences in the participant's experience with group ANC. In order to describe the experience of group ANC in an easy to understand manner, qualitative description, where the results are kept as close to the words of the participants, was used [14]. The use of an audit trail composed of methodological and analytical documentation and validation with other research team members was used to achieve validity [15].

## Ethics statement

The study aims and design were developed in collaboration with community members of the research team. The study was approved by the Ethical Review Committee of the Ghana Health Service and the University of Michigan Institutional Review Board (HUM00161464).

With IRB approval, oral consent was obtained from participants due high rates of illiteracy among Ghanaian women. The informed consent document was read aloud individually to all potential participants in their local language by Ghanaian research assistants. Using a teach-back method to confirm comprehension, the RAs then asked potential participants questions to ensure understanding of the research process and informed consent document, inviting questions until the information was clear. A health facility staff member signed that they were present while the benefits, risks, and procedures were read to the participant, that all questions were answered, and that women voluntarily agreed to take part in the research. For the midwives participating in the interviews, written consent was obtained in English.

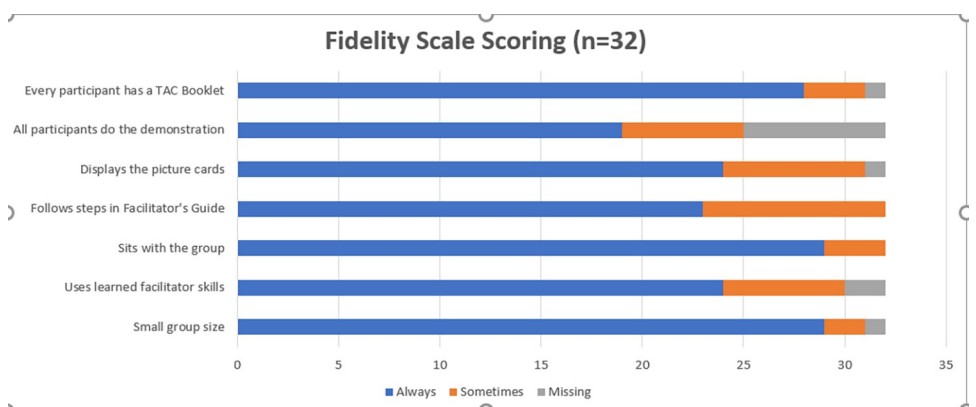

**Fig 1. Fidelity scale scoring.**

## Results

### Checklists

The Fidelity Checklist consists of 7 items intended to assess whether the facilitator uses techniques intended to encourage group dynamics and discussion. A total of 32 Fidelity Checklist Scales were completed by the champion trainers as they observed group ANC meetings among the intervention sites. The majority of the items were ranked as "Always" (177/224 responses), "Sometimes" (36/224 responses) with 11 missing responses. None of the responses were marked "Never". The item that scored the lowest with the most "Sometimes" vs. "Always" was item 4 "Follows the steps as written in the Facilitators Guide" (Fig 1). Examples of comments included on the Fidelity Checklist forms by the champion trainers during their observations included *"the meeting was very interactive"*, *"midwives should review the guide well before group meetings"* and *"ask open-ended questions to empower pregnant women to talk"*. The champion trainers reviewed the results with the midwife facilitators to provide timely feedback.

The Learning Methods Checklist with 16 items identifies which steps may have been missed in the meeting format. In most cases, observers indicated that all steps were followed, with only two observations indicating that the midwife did not "Go through the what and why" step of explaining what actions are needed when a problem arises and why those actions are taken. However, some checklists included comments that reminders were given by the observers, such as to *"Review the previous meeting"*. Comments also indicated challenges in time management such as *"Meeting started an hour late"* and *"Meeting was delayed"*. Resource management was also an issue, *"BP monitors were not charged"* at one facility and another comment was *"Not all resources were ready to start meeting."*

### Tracking logs

There was a total of 70 groups tracked across the 7 intervention sites between August 2019 and April 2022 (Fig 2). A total of 622 pregnant women were documented in the tracking logs. Of these participants, the majority (74%) attended more than half of the meetings, with 32% attending all 8 group ANC meetings. There were 11% who attended only one or two meetings, typically attending the first one or two Group ANC meetings and then dropping out. This might suggest they either miscarried or moved away, yet it could indicate dissatisfaction with the group care model. However, such data were not collected as most of these women were lost to follow up.

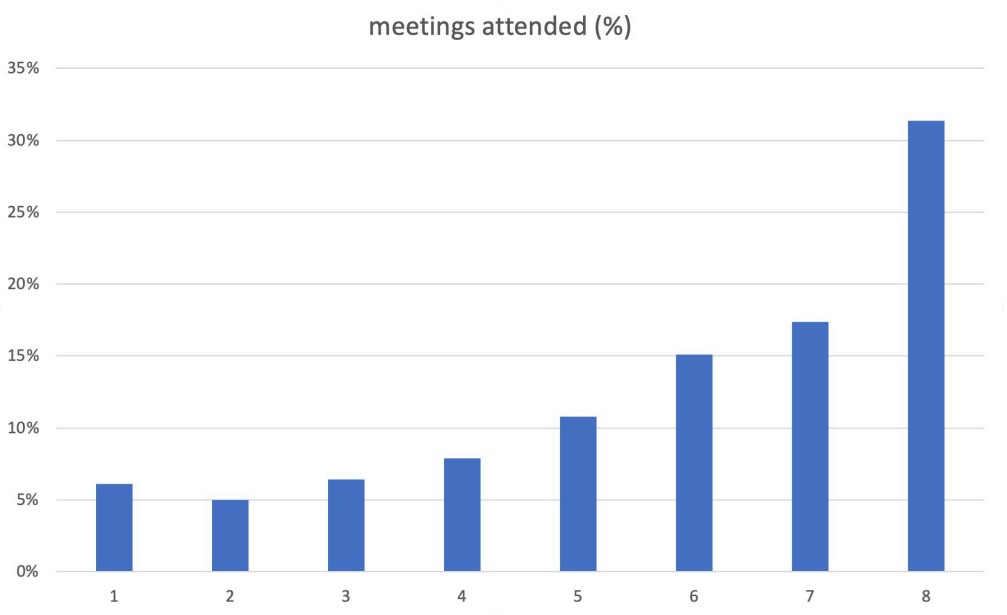

**Fig 2. Tracking logs.**

## Interviews with midwives

One-to-one interviews using a semi-structured interview guide were conducted by research team members with group ANC midwives at each of the 7 intervention sites (N = 7). Three themes were identified: 1) information sharing, 2) sense of community, and 3) time management challenges.

**Information sharing.** Midwives identified that they enjoyed sharing information with participants in a setting that offered the opportunity for group participation *"because this one is in a small group, they are able to feel free and to share experiences"* (Site 1 midwife) and that it also allowed time to discuss the "why" versus just the "what". *"I think the pregnant women appreciate knowing why we do something like take a blood pressure"* (Site 2 midwife). Midwives indicated that having a step-by-step guide for each meeting was very useful in preparing for the meetings and in keeping the meetings running smoothly, indicating it was also a source for the midwives to gain additional knowledge: *"We also get knowledge from it."* (Site 2 midwife). The midwives expressed optimism that group ANC would have long term educational benefits. *"It's going to go a long way to help our women understand the education we give them and the care and what they are supposed to do themselves during the pregnancy, labour and after delivery".* (Site 1 midwife).

**Sense of community.** Midwives felt that group ANC was beneficial in creating a sense of community among the women where information and support was coming from other women and not just from the midwife *"Because [group ANC] is a small group they are able to feel free and to share experiences so this. . .is much better than before."* (Site 1 midwife). Group ANC instilled trust between the woman and midwife because there was a consistent midwife in the group rather than the women seeing a different midwife each time

**Time management challenges.** Midwives indicated that starting on time was a challenge because not all of the women were there on time, and they needed to decide whether to finish all the material or end on time. They expressed optimism that time management would get easier with subsequent meetings: *"We are hoping that our next meeting things will be a bit faster then we minimize the number of hours or minutes we used today."* (Site 3 midwife).

## Focus groups with participants

A total of 10 focus group discussions with a total of 92 participants (5–13 women per group) were conducted after the final group ANC meetings. One focus group was conducted at 4 intervention sites and 2 focus groups were conducted at 3 of the sites. Some women in the groups had already delivered and attended with their newborns. Focus groups lasted approximately 30 minutes. Participants were overwhelmingly positive about their participation in group ANC and hoped it would continue. Qualitative, thematic analysis of focus group data revealed that participant experiences resulted in three themes identical to those in the midwifery interviews: 1) sense of community, 2) information sharing, 3) time management challenges, and an additional theme unique to the pregnant women; 4) disconnect.

**Sense of community.** Some participants expressed initial reluctance to participate that was quickly overcome. *"On the first day it was scary. . .but after I got involved, I realized it's comfortable and good and it is helping me too"* (Site 4 participant). Overwhelmingly, participants expressed they were happy they joined and were proud of their participation. As one participant said after the last group ANC visit, *"Everyone has become a graduate in pregnancy"* (Site 3 participant).

Participants in each group expressed their happiness that the midwives demonstrated respectful maternity care. *"You play with us, you make us happy, and you also teach us well- . . .we tend to forget some burdens we bring from home"* (Site 4 participant). They appreciated being treated with kindness and patience by the midwives who gave out their phone numbers, calling, and even visiting participants in their homes. *"The way the midwives treat them (us) it shows that they have love for the pregnant woman"* (Site 3 participant).Women felt a sense of connection with the midwives. *"I have never come across midwives like these. . .I feel they already knew us"* (Site 7 participant).

Women came back after their babies were born to share with others. They expressed a desire to continue group ANC so other women could participate. *"When I went to deliver my first born, I didn't get this kind of treatment. . .I want you to continue with it so that everyone will benefit to help all women who give birth"* (Site 5 participant). Participants asked that the meetings continue through the postnatal period. Some suggested they be involved in teaching others. *"I hope that there are other pregnant women who also need the education. . .We could come and also teach them what we did that helped us"* (Site 2 participant).

Participants enjoyed meeting new friends and relied on others for support. *"I am happy because you have brought togetherness and love between us and that makes me happy"* (Site 4 participant). Some even suggested having a fund for those with more financial need.

**Information sharing.** Participants were able to recall all topics covered. What is evident from the data is that it was important to participants that the *why* was included. *"At first you think it is normal, but due to the teaching you will know that the sickness you have is not normal and why you need to take the medicine"* (Site 3 participant). Knowledge helped to quell fears and dispel myths. *"It has taken away the fear we used to have when pregnant"* (Site 4 participant). Participants discussed learning about the importance of preparation for birth–specifically transportation, supplies and saving money for emergencies.

Participants expressed enthusiasm for the group ANC model of care where participants check one another's blood pressure and weights with the supervision of the midwife, expressing feelings of empowerment and agency. *"I was excited because. . .I check my BP and weigh myself"* (Site 2 participant). They gained confidence to ask questions of care providers. *"We were able to understand that when you come and they check you, you can ask how it is"* (Site 4 participant).

Participants in group ANC are given a booklet to take home as a helpful reminder of things they learn in the meetings. *"It's still helping because when I see some signs and I don't*

*understand I open the book to check to see if it's a danger sign or it's normal"* (Site 5 participant). Participants used the booklet to share the information with friends and family members which helped validate the information as correct. *"Sometimes when I go and tell my husband that we learned these things, at times he doesn't believe me, but I show him the book"* (Site 5 participant). Participants recalled the topics regarding partner communication and suggested that partners be encouraged to attend some meetings. *"They taught us how we should communicate with our partners. . .they taught us how to talk to our partners so that they can understand and support us"* (Site 5 participant).

Women who had experienced traditional ANC in prior pregnancies indicated that group ANC was preferred. Participants expressed greater knowledge acquisition and feelings of support during group ANC. *"This will be my 5th child. There are a lot of things I do not know about the previous 4 pregnancies but now. . .I will remember what we were taught here"* (Site 1 participant).

**Time management challenges.**   Similar to the interviews with midwives, this issue was mentioned in every group. Participants shared that it was hard for them to arrive on time. *"Looking at the time scheduled for us, it is difficult for some of us to access vehicles"* (Site 6 participant). Others expressed frustration that they were there on time and had to wait. *"My problem was with the time, we come early hoping that we can go early but when we come for the meeting we stay for long"* (Site 2 participant). At times it was the midwives who did not come on time. *"Make sure they come on time and also the (midwives) should also come on time"* (Site 1 participant).

**Disconnect.**   At times participants who had already given birth saw a disconnect between the group ANC teaching and the birth experience. In other words, participants were empowered to advocate but on the maternity ward they were not always treated with respect. One participant recalled being discounted by the staff: *"I told them that the baby was coming, they told me that it is not yet time and that I am not the one to tell them the time to check me"* (Site 1 participant). There were stories of disrespectful maternity care: *"They kept inserting their hands inside my vagina and in fact I was not happy at all"* (Site 1 participant).

## Discussion

Maternal and infant mortality in LMICs remains unacceptably high and women and infants that survive pregnancy and childbirth often suffer long term sequalae. Often, the traditional approach to ANC provides little in the way of education and counseling about health-promoting behaviors. To provide the most benefit, ANC must provide quality care that is respectful and valued by both the provider and the patient.

Group ANC models of care have been implemented successfully in LMIC countries with promising, but not consistent outcomes [16–19]. While group ANC in Rwanda did not result in a difference in gestational age at birth or number of ANC visits attended [17], participants felt they had increased knowledge and support and a closer relationship with their providers [20]. A study in Malawi found that women in the group ANC arm of the study were more likely to receive comprehensive services such as blood pressure screening at each visit and demonstrated increased knowledge of health-related topics than women assigned to individual antenatal care [18]. A pilot study conducted in Malawi and Tanzania found that participants in group ANC were better able to communicate with their partners and their partners were more likely to be tested for HIV [21]. Preliminary research in Ghana demonstrated an increase in health literacy or more specifically, how to recognize and prevent problems, when to access care, breast feeding and family planning methods [22]. Moreover, evidence from Kenya suggests group ANC may be associated with an increase in facility-based births as well as family preparedness for birth [23].

Feasibility of implementing group ANC has been explored previously and our results are similar [24, 25]. CenteringPregnancy was adapted and implemented in Mexico and found to be feasible and acceptable to both the health care professionals and pregnant women [24]. Adequate space for the group meetings and recruitment were the biggest challenges in this context [24]. A generic model of group ANC was developed by researchers in India, using elements from CenteringPregnancy and Home-Based Life Saving Skills [25]. Feasibility and acceptability were explored qualitatively by demonstrating one meeting to key informants (maternity care providers, administrators and pregnant women). Participants were supportive of the model of care, finding it more comprehensive, educational, and empowering vs. traditional care [25].

## Facilitators and barriers

**System.** Training adequate numbers of facilitators in this model of group ANC can be time consuming and costly. Using a train-the-trainer model of implementation, we were able to train champion trainers who then trained midwives at the intervention sites which increased efficiency. Training workshops occurred in small groups similar to the model of group ANC meetings so they could practice the facilitation techniques.

A concern expressed by the participants as well as the midwives was that of time management. Midwives stated that often women did not come on time. Some participants were frustrated when they came on time and the meeting did not start at the allotted time while others had issues with being on time due to transportation issues. Strategies for time management were addressed in "reminder" messages sent to the midwife facilitators.

**Midwives.** The midwives were able to administer the group ANC curricula with fidelity to the format using the techniques they learned and practiced in the training. The midwives that were interviewed all expressed a preference for group ANC over the standard, individual care model, feeling like it resulted in more education and a stronger sense of community. Each midwife was given a Facilitators Guide which they indicated helped them to cover everything in each meeting. The majority of midwives followed the steps of the ANC visit according to the Facilitators Guide, however there were some components that needed to be reinforced. This included the midwife asking open ended questions to facilitate discussion which may be contrary to the traditional educational delivery approach.

In the majority of cases, the midwives conducted the meetings in accordance with the group ANC model of care. The midwives delivered the content intended for that meeting and the observers noted that the pregnant women were encouraged to participate in group discussions. At times, activities such as demonstrations were omitted due to time constraints. The midwives utilized the picture cards and the Facilitators Guide consistently.

In Ghana, it is common for midwives to either work in the antenatal or intrapartum area and not in both. Because of that, some participants who gave birth before the final meeting focus group session described a disconnect between what they learned in the meetings and the care they received at the time of birth. Future implementation of group ANC should include training and information for midwives attending births as well as those in the ANC clinics.

**Pregnant women.** The research team was able to recruit pregnant participants willing to enroll in the study and participate in group ANC despite some delay due to the COVID pandemic. Of the participants, the majority attended four or more of the 8 meetings. One approach to encouraging attendance was reminder texts and phone calls by the midwives, although reasons for non-attendance were not collected. Given that the traditional model of ANC in Ghana included 4 visits, attendance at 8 visits by more than 1/3 of the women is encouraging. Women who participated in group ANC voiced enthusiasm for the model of

care and wished for it to continue. Having the opportunity to tell stories, voice opinions and ask questions in the group gave them a sense of empowerment as did self-checking blood pressures and weights. They enjoyed having materials to take home to share with friends and family. One of the most salient findings was that group ANC created a sense of community and also a connection to the midwives that they did not experience with prior antenatal care.

## Limitations

Implementation of the group ANC intervention and the ensuing process evaluation were impacted by the pandemic, as described above. However, the research team in Ghana and the midwife facilitators were committed to the program success. Fortunately, most of the group meetings were held outside which limited risk of COVID transmission and participants and midwives were required to wear masks during the meetings. While attempts were made by the midwives and research assistants to reach who stopped attending meetings, we do not have data on reasons for discontinuation and can only speculate that while some miscarried or moved, others may have not enjoyed or been dissatisfied with group antenatal care.

Process evaluations are increasingly incorporated into health intervention research, providing value by documenting the barriers and facilitators of the intervention components [26]. Our process evaluation was designed to monitor and document fidelity to the core components of the group ANC intervention. As noted, challenges were faced in time management, yet overall acceptability by midwives and women as well as fidelity to the group ANC model was high. Group ANC was preferable to those who have given birth prior to this pregnancy which highlights the value of community and underscores the need to keep content updated to the newest available information.

## Conclusion

Significant shifts in clinical practice do not happen easily or quickly. Valid, reproducible evidence must exist for practice guidelines to change. Implementation of group ANC in this context was not without challenges, one of which was time management, however our findings contribute to the growing evidence that group ANC in LMICs is safe, feasible and that women find it beneficial. We advocate for future research focusing on the scale-up of group ANC in LMICs to address the unacceptable high rates of maternal and newborn mortality.

## Supporting information

**S1 Checklist. Fidelity checklist.**
(DOCX)

**S2 Checklist. Learning methods checklist.**
(DOCX)

**S1 File. Tracking log.**
(DOCX)

**S2 File. Consolidated criteria for reporting qualitative research (COREQ) checklist.**
(DOCX)

**S3 File. PLOS ONE questionnaire on inclusivity.**
(DOCX)

## Author Contributions

**Conceptualization:** Vida Kukula, Cheryl Moyer, John Williams, Jody Lori.

**Data curation:** Vida Kukula, Veronica Apetorgbor, Elizabeth Awini, Georgina Badu-Gyan, Nancy Lockhart.

**Formal analysis:** Ruth Zielinski, Cheryl Moyer, Nancy Lockhart, Jody Lori.

**Funding acquisition:** Cheryl Moyer, Jody Lori.

**Investigation:** Ruth Zielinski, Vida Kukula, Cheryl Moyer, Jody Lori.

**Methodology:** Ruth Zielinski, Vida Kukula, Cheryl Moyer, Jody Lori.

**Project administration:** Vida Kukula, Veronica Apetorgbor, Elizabeth Awini, Cheryl Moyer, Georgina Badu-Gyan, John Williams, Nancy Lockhart.

**Resources:** Vida Kukula, Nancy Lockhart.

**Software:** Nancy Lockhart.

**Supervision:** Vida Kukula, Veronica Apetorgbor, Elizabeth Awini, Georgina Badu-Gyan, John Williams, Nancy Lockhart, Jody Lori.

**Validation:** Ruth Zielinski, Vida Kukula, Veronica Apetorgbor, Elizabeth Awini, Cheryl Moyer, Jody Lori.

**Visualization:** Jody Lori.

**Writing – original draft:** Ruth Zielinski, Vida Kukula, John Williams.

**Writing – review & editing:** Ruth Zielinski, Vida Kukula, Veronica Apetorgbor, Elizabeth Awini, Cheryl Moyer, Georgina Badu-Gyan, John Williams, Nancy Lockhart, Jody Lori.

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
