## [Decision Letter · Decision Letter 0]

18 Aug 2022

PONE-D-22-17328“With Group Antenatal Care, Pregnant Women Know They are Not Alone”: The Process Evaluation of a Randomized Controlled Trial of Group Antenatal Care in GhanaPLOS ONE

Dear Dr. Zielinski,

Thank you for submitting your manuscript to PLOS ONE. After careful consideration, we feel that it has merit but does not fully meet PLOS ONE’s publication criteria as it currently stands. Therefore, we invite you to submit a revised version of the manuscript that addresses the points raised during the review process.

We look forward to receiving your revised manuscript.

Kind regards,

Godwin Otuodichinma Akaba, MBBS,MSc,MPH,FWACS

Academic Editor

PLOS ONE

Journal Requirements:

4. We note that you refer to a Clinical Trial (.........) in your manuscript. Please include the name of the registry and the registration number (e.g. ISRCTN or ClinicalTrials.gov) in your manuscript. If the results have been previously reported, please provide a reference to the publication.

" Author JL received the award

National Institute of Health/National Institute of Nursing Research

https://www.ninr.nih.gov/

The funders had no role in study design, data collection and analysis, decision to publish, or preparation of the manuscript"

7. Please ensure that you refer to Figure 2 in your text as, if accepted, production will need this reference to link the reader to the figure.

Additional Editor Comments:

This was a manuscript that describes the process evaluation for a cluster randomized trial that compared group antenatal care and the traditional model of antenatal care. The purpose of the process evaluation was to identify and document patient, provider, and system barriers and facilitators to program implementation.

Process evaluations usually are carried out examine the extent to which a program is operating as intended by assessing progress at attaining desired goals and objectives with a view of sustaining progress made and realigning focus when barriers to attaining program objectives are encountered.

The manuscript has been well written, however there are areas of the manuscript that needs to be revised to improve the quality of the article.

Background

Page 2, line 27: Interviews-Please change to interviewed

Page 3, lines 62: Antenatal care (ANC) defined by the World Health Organization (5)

Comment: Consider removing the reference (5) since this is still reflected at the end of the whole statement.

Page 3, line 64: Please delete P.1 in the reference in bracket.

Page 4, line 82: Details of the project have been described elsewhere (7).

Comment: The study noted that details of the project have been described elsewhere. However, the quoted reference cannot be accessed thereby making review of the detailed project protocol impossible. This reference should be properly cited. Additionally, it is not completely in order that a funded cluster randomized trail protocol was not registered or published before commencement of the study. A more detailed description of what was done should be included in this manuscript in the absence of a link for readers to assess the detailed protocol. This will enable easy understanding of the entire project from which the process evaluation is being conducted.

Methods

Comment: It may be worthwhile to explain the theoretical framework(s) considered for the conduct of the trial. This will help readers to interpret the process evaluation findings

Comment: This process evaluation did not mention or consider the other arm of the cluster randomized study. It is therefore not clear how the interpretation of findings at the end of the study would be done if this important arm was completely not mentioned in the process evaluation. This is a major flaw in the design of the process evaluation.

Results

Page 10, lines 201-205

Comment: One of the essences of process evaluation is to see whether the project is on track to deliver on intended goals as well as shape modifiable factors to help achieve the expected outcomes. It would have been appropriate to include opportunities to track this group of women as their non-attendance could be from dissatisfaction with how the project is being implemented. Their loss has closed an opportunity to learn and improve thereby diminishing the benefits of the process evaluation.

Page 12, lines 254-257:

Comment: If truly the participants were called and visited at home, the 11% lost to follow up should have been identified and interviewed on the possible reasons for their non-continuity with the ANC rather than speculating the reasons in the discussion section.

Page 14, line 293: Time management challenges

Comment: This barrier or challenge should be discussed in the discussion section on how to improve this before the end of the project

Page 15, line 303.

Comment: It will be appropriate to consider expanding on this in the discussion section. For example, will there be need to consider including recent advancements and information during the ANC sessions so that even those who have delivered before will look forward to learning something new?

Additional comment on the result: The qualitative component should please be reported in accordance with the COREQ guideline with attachment of same as a supplementary material. 

Discussion

Page 15, lines 310-324

Comment: These appeared like repetitions of what had been said in the background section. Preferably a statement or two on the essence of process evaluation and followed by the discussion of findings would suffice.

Lessons learnt particularly the challenges and obstacles to implementation have not been discussed. Hence losing a component of the essence of process evaluation.

The result and discussion should be revised to include the control arm of the study which an important 50% of the study design.

On a general note, the reviewers require that:

1.The results and discussion section should include the control arm of the cluster randomized trial

2.The reference of the study protocol being considered for review should be properly cited. However, if the link is not available for readers to assess, authors should include a brief description of what was done in the groups. 

3. The discussion section should highlight implications of the barriers and challenges to uptake of the group antenatal care and possible ways of addressing these barriers towards attaining the project goals.

Reviewers' comments:

Reviewer's Responses to Questions

**Comments to the Author**

1. Is the manuscript technically sound, and do the data support the conclusions?

Reviewer #1: Partly

Reviewer #2: Yes

2. Has the statistical analysis been performed appropriately and rigorously? 

Reviewer #1: No

Reviewer #2: Yes

3. Have the authors made all data underlying the findings in their manuscript fully available?

Reviewer #1: No

Reviewer #2: Yes

4. Is the manuscript presented in an intelligible fashion and written in standard English?

Reviewer #1: No

Reviewer #2: Yes

5. Review Comments to the Author

Reviewer #1: the clients are randomized into two groups (one received routine ANC and the other group ANC).Authors should provide the findings from the routine ANC group (primary outcomes) in the result section.

The discussion should be based on the availability of the data above.

Authors should use statistical analysis to compare the two models of ANC to make inference

Reviewer #2: The article demonstrated a process evaluation of a randomized controlled trial of group antenatal care in Ghana. The research design is interesting, and the research is generally well performed. The key uncertainties were identified and the most important questions to be addressed were selected. The researchers combined both quantitative and qualitative data on key process variables, however, there are some deficiencies throughout the manuscript that would need further clarification.

Specific comment

Title:

Line 1: ‘Are’ should be capitalized

Abstract

Lines 25/27/39: correct typographical errors. Authors may consider hiring a copyeditor.

Introduction

Line 64: clarify reference

Line 82: First sentence of line 82 may come at the end of the paragraph, while line 82-87 should be moved to line 78.

Line 82: correct spelling of ‘elsewhere’

Line 85: qualify care seeking eg care seeking behaviour

Line 95: may qualify study eg GRAND study

Line 99 to 103: should be paraphrased to reflect the aim of the study not results.

Methods

It is suggested to add subheadings and write them in order for example, Study setting and design, sample size, research team/data collection, data analysis (more details), ethics statement (approval no. should be included)

Line 108-117 can be under study setting and design, however, more context should be added.

Line 118-119: it is unclear who the process evaluators were and if they were separate from the outcome evaluators. If same, how do the authors ensure the results are not biased.

Line 118-165 should be under data collection and should be further elucidated

Line 121: Table 2 should not cross a page

Line 154: It is suggested the word ‘dose’ be defined in the introduction section as well as ‘fidelity’, being the key components of the process evaluation.

Results

A brief demographics of the participants (both midwives and women) included in the study may be provided

Line 178: correct typographical error

Line 180: the word ‘Never’ should be in parenthesis

Line 180-182: this is not what is reflected in Figure 1

Figure 1: ‘Never’ was on the scale and ‘Missing’ on the figure, there should be consistency.

Proper title and axes labels should be added.

Line 182: it is unclear who made the comments, you may wish to specify

Line 182: ‘midwives’ rather than ‘midwife’

Figure 2: Proper title and axes labels should be added.

Line 226: Paraphrase eg a sense of community or community partnership

Line 240: indicate the number of intervention sites the focus groups were held

Line 256: is this statement made by a pregnant woman? If yes, why the use of the word “them”

Line 313: cross-check

Line 320: clarify reference

Discussion

It is unclear if the discussion was in line with the aim of process evaluation. More context should be given about process evaluation including the data collection tools while relating the information to the GRAND study.

Barriers and facilitators to the implementation of group antenatal care identified from this process evaluation should be elucidated.

The answers to the questions listed in lines 128-131 should be discussed to describe the fidelity of the model while the results of the tracking logs should be discussed to describe the dose of the model.

Conclusion

Line 379-383 may be moved to the discussion section.

Please give a concise conclusion.

6. PLOS authors have the option to publish the peer review history of their article (what does this mean?). If published, this will include your full peer review and any attached files.

Reviewer #1: No

Reviewer #2: No

---

## [Author Response · Author response to Decision Letter 0]

4 Oct 2022

Editors Comments response

Thank you, the manuscript has been edited and revised to meet these requirements. 

Please include a complete copy of PLOS’ questionnaire on inclusivity in global research in your revised manuscript. Please upload a completed version of your questionnaire as Supporting Information when you resubmit your manuscript. Completed and uploaded as supporting information – thank you

3. Please provide additional details regarding participant consent. In the ethics statement in the Methods and online submission information, please ensure that you have specified what type you obtained (for instance, written or verbal, and if verbal, how it was documented and witnessed). If your study included minors, state whether you obtained consent from parents or guardians. If the need for consent was waived by the ethics committee, please include this information. Additional details have been included (page 7 line 125-133)

Once you have amended this/these statement(s) in the Methods section of the manuscript, please add the same text to the “Ethics Statement” field of the submission form (via “Edit Submission”). The same text was added to the “Ethics Statement” field of the submission

We note that you refer to a Clinical Trial (.........) in your manuscript. Please include the name of the registry and the registration number (e.g. ISRCTN or ClinicalTrials.gov) in your manuscript. If the results have been previously reported, please provide a reference to the publication. Clinical trial # NCT 04033003 added on page 6 line 124 of the manuscript. 

. Thank you for stating the following financial disclosure:

" Author JL received the award National Institute of Health/National Institute of Nursing Research https://www.ninr.nih.gov/The funders had no role in study design, data collection and analysis, decision to publish, or preparation of the manuscript"

Please include your amended statements within your cover letter; we will change the online submission form on your behalf. Thank you, our response has been included in the updated cover letter

We note that you have stated that you will provide repository information for your data at acceptance. Should your manuscript be accepted for publication, we will hold it until you provide the relevant accession numbers or DOIs necessary to access your data. If you wish to make changes to your Data Availability statement, please describe these changes in your cover letter and we will update your Data Availability statement to reflect the information you provide. All repository data is compiled and will be uploaded to Deep Blue on acceptance of the manuscript, thank you. 

Please ensure that you refer to Figure 2 in your text as, if accepted, production will need this reference to link the reader to the figure. Reference was made to Figure 2 (page 12 line 238)

Please include captions for your Supporting Information files at the end of your manuscript, and update any in-text citations to match accordingly. Please see our Supporting Information guidelines for more information: http://journals.plos.org/plosone/s/supporting-information. The manuscript has been edited to include captions and in-text citations using the guidelines supplied, thank you. 

Page 2, line 27: Interviews-Please change to interviewed

Page 3, lines 62: Antenatal care (ANC) defined by the World Health Organization (5)

Comment: Consider removing the reference (5) since this is still reflected at the end of the whole statement.

Page 3, line 64: Please delete P.1 in the reference in bracket. Interviews changed to interviewed

(5) removed

P.1 deleted

Page 4, line 82: Details of the project have been described elsewhere (7). The study noted that details of the project have been described elsewhere. However, the quoted reference cannot be accessed thereby making review of the detailed project protocol impossible. This reference should be properly cited. Additionally, it is not completely in order that a funded cluster randomized trail protocol was not registered or published before commencement of the study. A more detailed description of what was done should be included in this manuscript in the absence of a link for readers to assess the detailed protocol. This will enable easy understanding of the entire project from which the process evaluation is being conducted. I apologize that this was not clear. The protocol has since been published and the reference has been updated (page 5 line 100). The protocol was registered and available at https://clinicaltrials.gov/ct2/show/NCT04033003?term=04033003&draw=2&rank=1

It may be worthwhile to explain the theoretical framework(s) considered for the conduct of the trial. This will help readers to interpret the process evaluation findings Reference has been made to the Protocol Paper text was added to this paper (p5 l99-100)

This process evaluation did not mention or consider the other arm of the cluster randomized study. It is therefore not clear how the interpretation of findings at the end of the study would be done if this important arm was completely not mentioned in the process evaluation. This is a major flaw in the design of the process evaluation. I apologize that the way it was written in the last draft was confusing. This was a process evaluation of the intervention (group antenatal care) rather than of the entire randomized study. We have added clarification of this throughout the paper. Thank you for helping us add clarity. To this end we also revised the title from “The process evaluation of a Group Antenatal Care Intervention” vs. Process evaluation of an RCT…”

Page 10, lines 201-205

Comment: One of the essences of process evaluation is to see whether the project is on track to deliver on intended goals as well as shape modifiable factors to help achieve the expected outcomes. It would have been appropriate to include opportunities to track this group of women as their non-attendance could be from dissatisfaction with how the project is being implemented. Their loss has closed an opportunity to learn and improve thereby diminishing the benefits of the process evaluation. This is definitely a challenge, attempts were made to follow up with women when they did not attend but we do not have data as to reasons for discontinuation. We agree, this would have been valuable information to have and include and have added this to the limitations of the study (page 20 line 424)

Page 12, lines 254-257:

Comment: If truly the participants were called and visited at home, the 11% lost to follow up should have been identified and interviewed on the possible reasons for their non-continuity with the ANC rather than speculating the reasons in the discussion section. That we did not collect reasons which was a limitation of the study. A point of clarification is that it was not part of the protocol call and/or visit, this was an observation made by the participants. 

Page 14, line 293: Time management challenges

Comment: This barrier or challenge should be discussed in the discussion section on how to improve this before the end of the project Thank you for this suggestion, a section was added to the discussion “Facilitators and barriers (page 18 line 377)

Page 15, line 303.

Comment: It will be appropriate to consider expanding on this in the discussion section. For example, will there be need to consider including recent advancements and information during the ANC sessions so that even those who have delivered before will look forward to learning something new? Interesting point, those who had given birth before expressed their preference for group ANC which highlights the value of not only the education but also community. Additional information added (page 20 line 433-435)

Additional comment on the result: The qualitative component should please be reported in accordance with the COREQ guideline with attachment of same as a supplementary material. Additional text regarding the qualitative component was added to the manuscript and a 32 item COREQ checklist was added to the supplementary material . 

Page 15, lines 310-324 These appeared like repetitions of what had been said in the background section. Preferably a statement or two on the essence of process evaluation and followed by the discussion of findings would suffice. Thank you for this observation, this content has been revised with some that was not repetition of the introduction moved there and the repetitious content was deleted. 

Lessons learnt particularly the challenges and obstacles to implementation have not been discussed. Hence losing a component of the essence of process evaluation. Thank you for this observation – a sub-section titled “Facilitators and barriers” has been added

The result and discussion should be revised to include the control arm of the study which an important 50% of the study design. I apologize for our lack of clarity that led to confusion on the part of all reviewers. This was a process evaluation of the implementation of the intervention not both arms of the study – We have made changes throughout the manuscript

The results and discussion section should include the control arm of the cluster randomized trial This was a process evaluation of the implementation of the intervention not both arms of the study – We have made changes throughout the manuscript and appreciate the confusion this caused

The reference of the study protocol being considered for review should be properly cited. However, if the link is not available for readers to assess, authors should include a brief description of what was done in the groups This has been updated to include the reference which is now published and available – thank you. 

The discussion section should highlight implications of the barriers and challenges to uptake of the group antenatal care and possible ways of addressing these barriers towards attaining the project goals. Thank you for this suggestion. A sub-section titled “Facilitators and Barriers” has been added. 

Reviewer 1 

the clients are randomized into two groups (one received routine ANC and the other group ANC).Authors should provide the findings from the routine ANC group (primary outcomes) in the result section. I apologize for our lack of clarity that led to confusion on the part of all reviewers. This was a process evaluation of the implementation of the intervention not both arms of the study – We have made changes throughout the manuscript

The discussion should be based on the availability of the data above. Apologies - See above

Authors should use statistical analysis to compare the two models of ANC to make inference Apologies - See above

Reviewer 2 

Title: Line 1: ‘Are’ should be capitalized Thank you for noticing this – the title has been edited to correct PLOS-ONE format

Title: Line 1: ‘Are’ should be capitalized Abstract

Lines 25/27/39: correct typographical errors. Authors may consider hiring a copyeditor. Thank you for your attention to these details, the corrections have been made.

Line 64: clarify reference

Line 82: First sentence of line 82 may come at the end of the paragraph, while line 82-87 should be moved to line 78.

Line 82: correct spelling of ‘elsewhere’ The corrections have been made, thank you

Line 85: qualify care seeking eg care seeking behaviour Edited to care seeking behavior, thank you

Line 95: may qualify study eg GRAND study Edited to GRAND study, thank you

Line 99 to 103: should be paraphrased to reflect the aim of the study not results. This was edited to reflet the aim (p 6 114-119)

It is suggested to add subheadings and write them in order for example, Study setting and design, sample size, research team/data collection, data analysis (more details), ethics statement (approval no. should be included) Thank you for this suggestion, subheadings were added, however they did not fit as well into the traditional sub-headings given that this was a process evaluation. 

Line 108-117 can be under study setting and design, however, more context should be added. These edits were made, thank you 

Line 118-119: it is unclear who the process evaluators were and if they were separate from the outcome evaluators. If same, how do the authors ensure the results are not biased. The champion trainers (who also conducted the training) conducted the process evaluation. The outcome evaluation was collected by the Research assistants. 

Line 118-165 should be under data collection and should be further elucidated This content was moved to data collection and some additional clarification was added

Line 121: Table 2 should not cross a page Corrected, thank you

Line 154: It is suggested the word ‘dose’ be defined in the introduction section as well as ‘fidelity’, being the key components of the process evaluation. Thank you for this suggestion, for clarity, rather than placing the definitions in the introduction, “dose” and “fidelity” were defined at the first point in which they were introduced in the manuscript. 

A brief demographics of the participants (both midwives and women) included in the study may be provided None for this group of participants

Line 178: correct typographical error

Line 180: the word ‘Never’ should be in parenthesis

Line 180-182: this is not what is reflected in Figure 1

Figure 1: ‘Never’ was on the scale and ‘Missing’ on the figure, there should be consistency.

Proper title and axes labels should be added.

Line 182: it is unclear who made the comments, you may wish to specify

Line 182: ‘midwives’ rather than ‘midwife’

Figure 2: Proper title and axes labels should be added.

Line 226: Paraphrase eg a sense of community or community partnership

Line 240: indicate the number of intervention sites the focus groups were held

Line 256: is this statement made by a pregnant woman? If yes, why the use of the word “them”

Line 313: cross-check

Line 320: clarify reference Typographical errors were corrected, thank you. 

“Never” was placed in quotations

Thank you for finding the discrepancy between the text and Figure 1. This was corrected in the manuscript. As there were no “never” responses it was not included on the figure

Thank you for bringing this lack of clarity to our attention. We clarified in the manuscript that the comments were added on the Fidelity Checklist sheets by the champion trainers during their observation of the group ANC meetings. 

“community” was edited to “a sense of community”

All intervention sites had at least 1 focus group and 2 were conducted at 3 of the sites. This information has been added to the manuscript.

Line 313 – this content was moved to the introduction and seems to be consistent across all articles I could find that the pandemic has resulted in worsening outcomes for mothers and neonates. 

We recognize that the use of “them” is confusing, it was how it was translated from Twi to English. We added “us” in parenthesis for clarity.

Line 320 – this content was removed from the manuscript in response to an earlier reviewer suggestion

It is unclear if the discussion was in line with the aim of process evaluation. More context should be given about process evaluation including the data collection tools while relating the information to the GRAND study. The discussion section has undergone revision with these suggestions in mind

Barriers and facilitators to the implementation of group antenatal care identified from this process evaluation should be elucidated. A sub-section titled “facilitators and barriers” has been added

The answers to the questions listed in lines 128-131 should be discussed to describe the fidelity of the model while the results of the tracking logs should be discussed to describe the dose of the model. This has been incorporated into the discussion section

Line 379-383 may be moved to the discussion section. This content was moved to the discussion section

Please give a concise conclusion. The conclusion was edited for conciseness.

---

## [Decision Letter · Decision Letter 1]

11 Jan 2023

PONE-D-22-17328R1

“With group antenatal care, pregnant women know they are not alone”: The process evaluation of a group antenatal care intervention in Ghana.

PLOS ONE

Dear Dr. Zielinski,

Thank you for submitting your manuscript to PLOS ONE. After careful consideration, we feel that it has merit but does not fully meet PLOS ONE’s publication criteria as it currently stands. Therefore, we invite you to submit a revised version of the manuscript that addresses the points raised during the review process.

In addition to Reviewer #1's comment below, kindly address the following comments:

In complying with the journal standards, I will suggest the authors clearly articulate and describe the study population and settings, sample size calculation, how the sample size for each study components were derived, and the data collection procedure for these components. It is unclear the way these sections currently reads. For instance lines 237-243 (page 12 of the manuscript) include useful information that will help readers understand the study population and study setting, yet not described in the methods. It is unclear what these intervention sites are.  

For easy of understanding and replication of the study, I will suggest the methods section follow this order: Study design, study setting, study population, sampling, data collection process, data analysis and ethical approval. See example below:

Pilgrim N, Jani N, Mathur S, Kahabuka C, Saria V, Makyao N, et al. Provider perspectives on PrEP for adolescent girls and young women in Tanzania: The role of provider biases and quality of care. PloS one. 2018; 13(4):e0196280. https://doi.org/10.1371/journal.pone.0196280 PMID: 29702659

I also struggled to understand the data collection processes. The authors claimed they conducted both in-depth interviews (IDIs) and focus group discussions (FGDs) but did not clearly define the processes they followed in conducting these interviews. It is unclear how the data was collected and analyzed. For instance, how many IDIs and FGDs were conducted; how many participants were included in each FGDs. Also, they need to state what measures were used to determine the inclusion criteria for participating in the IDIs and FDGs. More so, the demographic characteristics of the participants should be include.

The qualitative analysis plan is vague. It is unclear what types of analyses was performed for both the IDIs and FGDs. For instance, the authors mentioned in lines 207-213 (page 10-11), that “Qualitative data were analyzed thematically…”, but failed to describe in detail how the data from the participants were analyzed. Was this done inductively and deductively or both? These are very useful details and yet not discussed in depth. I would like to see a clear analysis plan in the methods. This should clearly be stated for easy replication of the study.

I found it very difficult to understand and follow the results section due to the style adopted. The way the qualitative data is presented makes it difficult to read. All quotes should be referenced using codes to protect the identity of the participants. I will suggest using codes such as the ones used in Pilgrim N, Jani N, Mathur S, Kahabuka C, Saria V, Makyao N, et al. Provider perspectives on PrEP for adolescent girls and young women in Tanzania: The role of provider biases and quality of care. PloS one. 2018; 13(4):e0196280. https://doi.org/10.1371/journal.pone.0196280 PMID: 29702659. 

Also, lines 114-115 on page 6 of the manuscript states that "The purpose of this process evaluation was to identify and document patient, provider, and system barriers and facilitators...", however, the results and discussions did not present the barriers and facilitators differentiating these levels. I will suggest that the findings are categorized according to these levels. For example which are patient, provider barrier and/or facilitators, and which are systems barriers and/or facilitators. 

We look forward to receiving your revised manuscript.

Kind regards,

Edward Nicol, PhD

Academic Editor

PLOS ONE

Reviewers' comments:

Reviewer's Responses to Questions

**Comments to the Author**

1. If the authors have adequately addressed your comments raised in a previous round of review and you feel that this manuscript is now acceptable for publication, you may indicate that here to bypass the “Comments to the Author” section, enter your conflict of interest statement in the “Confidential to Editor” section, and submit your "Accept" recommendation.

Reviewer #1: All comments have been addressed

Reviewer #2: All comments have been addressed

2. Is the manuscript technically sound, and do the data support the conclusions?

Reviewer #1: Yes

Reviewer #2: Yes

3. Has the statistical analysis been performed appropriately and rigorously? 

Reviewer #1: I Don't Know

Reviewer #2: Yes

4. Have the authors made all data underlying the findings in their manuscript fully available?

Reviewer #1: Yes

Reviewer #2: Yes

5. Is the manuscript presented in an intelligible fashion and written in standard English?

Reviewer #1: Yes

Reviewer #2: Yes

6. Review Comments to the Author

Reviewer #1: the author should address minor issues like punctuations, spaces between words and minor spelling mistakes especially from the quoted statements from the designed proforma.

Reviewer #2: The authors have made all corrections I suggested and clarified the questions I raised in my previous review.

7. PLOS authors have the option to publish the peer review history of their article (what does this mean?). If published, this will include your full peer review and any attached files.

Reviewer #1: **Yes: **Ayyuba RABIU

Reviewer #2: No

---

## [Author Response · Author response to Decision Letter 1]

20 Jun 2023

Suggested revision: Clearly articulate and describe the study population and settings, sample size calculation, how the sample size for each study components were derived, and the data collection procedure for these components 

Author’s response: Thank you for this suggestion. We added content to better describe the study participants, setting and data collection procedures. Since this is a process evaluation, sample size calculations were not included in this report. The study protocol is referenced in the text.

Suggested revision: The authors claimed they conducted both in-depth interviews (IDIs) and focus group discussions (FGDs) but did not clearly define the processes they followed in conducting these interviews. It is unclear how the data was collected and analyzed. For instance, how many IDIs and FGDs were conducted; how many participants were included in each FGDs. Also, they need to state what measures were used to determine the inclusion criteria for participating in the IDIs and FDGs. More so, the demographic characteristics of the participants should be include. 

Authors response: More information has been added regarding how the interviews and focus groups were conducted. Inclusion criteria has been added. Regarding demographic characteristics. This data was collected for the group ANC outcome evaluation and will be included in those reports, however for the process evaluation the focus group data were analyzed and presented without using any identifiers such as demographic data or facility name. We did not collect demographic data for the midwife facilitator participants. 

Suggested revision: It is unclear the way these sections currently reads. For instance lines 237-243 (page 12 of the manuscript) include useful information that will help readers understand the study population and study setting, yet not described in the methods. It is unclear what these intervention sites are. Authors response: Additional content has been added that describes the study population, setting and intervention sites. 

Suggested revision: For easy of understanding and replication of the study, I will suggest the methods section follow this order: Study design, study setting, study population, sampling, data collection process, data analysis and ethical approval. See example below: 

Authors response: The methods section has been modified to follow the order suggested

Suggested revision: The qualitative analysis plan is vague. It is unclear what types of analyses was performed for both the IDIs and FGDs. For instance, the authors mentioned in lines 207-213 (page 10-11), that “Qualitative data were analyzed thematically…”, but failed to describe in detail how the data from the participants were analyzed. Was this done inductively and deductively or both? These are very useful details and yet not discussed in depth. I would like to see a clear analysis plan in the methods. This should clearly be stated for easy replication of the study. 

Authors response: We agree that the qualitative analysis was not well described in the manuscript. We used a qualitative comparison design but did not identify that in the manuscript. We added the steps taken in the analysis and added a reference for qualitative comparison designs. 

Suggested revision: I found it very difficult to understand and follow the results section due to the style adopted. The way the qualitative data is presented makes it difficult to read. All quotes should be referenced using codes to protect the identity of the participants. I will suggest using codes such as the ones used in Pilgrim N, Jani N, Mathur S, Kahabuka C, Saria V, Makyao N, et al. Provider perspectives on PrEP for adolescent girls and young women in Tanzania: The role of provider biases and quality of care. PloS one. 2018; 13(4):e0196280. https://doi.org/10.1371/journal.pone.0196280 PMID: 29702659. 

Authors response: As this was the report of a process evaluation, we did not collect identifiers for the participants and thus did not need to use codes to protect the identity of the participants. Regarding the presentation of the qualitative data, we used a presentation style similar to a process evaluation published in Plos One in 2021 that seems a better fit for this manuscript as it is also a process evaluation: Arthur K, Christofides N, Nelson G. Process evaluation of a pre-adolescent transdisciplinary health intervention for inter-generational outcomes. PLoS One. 2021 Dec 23;16(12):e0261632. doi: 10.1371/journal.pone.0261632. PMID: 34941911; PMCID: PMC8699635.

---

## [Editor Report · Decision Letter 2]

18 Jul 2023

PONE-D-22-17328R2“With group antenatal care, pregnant women know they are not alone”: The process evaluation of a group antenatal care intervention in Ghana.PLOS ONE

Dear Dr. Zielinski,

Thank you for submitting your manuscript to PLOS ONE. After careful consideration, we feel that it has merit but does not fully meet PLOS ONE’s publication criteria as it currently stands. Therefore, we invite you to submit a revised version of the manuscript that addresses the points raised during the review process.

To ensure the Editors are able to recommend that your revised manuscript is accepted, please pay careful attention to each of the comments below. This way, we can avoid future rounds of clarifications and revisions.

Constructive comments were proposed to the authors to improve the structure of the paper, however, some of these comments were not addressed. Specifically:

I found it difficult to understand and follow the results section due to the style adopted. The way the qualitative data is presented makes it difficult to read. All quotes should be referenced using codes to protect the identity of the participants. I will suggest using codes such as those used in Pilgrim et al 2018... The paper cited by the authors, in their rebuttal letter, also used the same reporting style, referencing the quotes with codes (Quintile 1 educator, Quintile 2 educator etc). See page 10 under **Quality of implementation** "Another educator encouraged participation by directing her questions to learners who did not volunteer to answer questions. *“Children don’t answer questions in my class except for the ones that speak excellent English. Other children sit quietly and try not to answer because of their lack of confidence to speak English, so when I teach I ask questions to the quieter children”* (Quintile 1 educator). See also page 12 under **Acceptability ***“The workbook and information in it, that was delivered, speak directly to the syllabus. I will use the workbooks again next year to make things easier to teach”* (Quintile 4 educator).Also, lines 114-115 on page 6 of the initial manuscript states that "The purpose of this process evaluation was to identify and document patient, provider, and system barriers and facilitators...", however, the results and discussions did not present the barriers and facilitators, differentiating these levels. i will suggest the findings are categorized according to these levels. For example, which of these results are patient, provider barriers and/or facilitators, and which are systems barriers and/or facilitators.  

We look forward to receiving your revised manuscript.

Kind regards,

Edward Nicol, PhD

Academic Editor

PLOS ONE

---

## [Author Response · Author response to Decision Letter 2]

23 Aug 2023

Reviewer/editor comment: All quotes should be referenced using codes to protect the identity of the participants 

Author response: Quotes were referenced using codes as suggested (Site # midwife) (Site # participant)

Reviewer/editor comment: lines 114-115 on page 6 of the initial manuscript states that "The purpose of this process evaluation was to identify and document patient, provider, and system barriers and facilitators...", however, the results and discussions did not present the barriers and facilitators, differentiating these levels. i will suggest the findings are categorized according to these levels. 

Author response: Thank you for this suggestion. Under the heading “Barriers & facilitators” is now subheadings “System”, “Midwife” “Pregnant Women”. The content was adjusted accordingly

---

## [Editor Report · Decision Letter 3]

7 Sep 2023

“With group antenatal care, pregnant women know they are not alone”: The process evaluation of a group antenatal care intervention in Ghana.

PONE-D-22-17328R3

Dear Dr. Zielinski,

We’re pleased to inform you that your manuscript has been judged scientifically suitable for publication and will be formally accepted for publication once it meets all outstanding technical requirements.

Kind regards,

Edward Nicol, PhD

Academic Editor

PLOS ONE
---

## [Editor Report · Acceptance letter]

30 Oct 2023

PONE-D-22-17328R3 

“With group antenatal care, pregnant women know they are not alone”: The process evaluation of a group antenatal care intervention in Ghana 

Dear Dr. Zielinski:

I'm pleased to inform you that your manuscript has been deemed suitable for publication in PLOS ONE. Congratulations! Your manuscript is now with our production department. 

Kind regards, 

on behalf of

Dr. Edward Nicol 

Academic Editor

PLOS ONE